# Rolling and Rolling-Sliding Contact Fatigue Failure Mechanisms in 32 CrMoV 13 Nitrided Steel—An Experimental Study

Luís Coelho [1,2,*], António C. Batista [2], João Paulo Nobre [2] and Maria José Marques [2,3]

1    School of Technology and Management, Polytechnic of Leiria, 2411-901 Leiria, Portugal
2    CFisUC, Department of Physics, University of Coimbra, 3004-516 Coimbra, Portugal; castanhola@uc.pt (A.C.B.); joao.nobre@dem.uc.pt (J.P.N.); mjvaz@fe.up.pt (M.J.M.)
3    Department of Physics Engineering, Faculty of Engineering, University of Porto, 4200-465 Porto, Portugal
*    Correspondence: luis.coelho@ipleiria.pt

**Abstract:** The aim of this work is to characterize the rolling and rolling-sliding contact fatigue failure mechanisms on the 32CrMoV13 nitrided steel. During rolling contact fatigue tests (RCF), two general features were observed: specimens presenting short lives and rough and sharpened spalling damage and specimens presenting long lives and only microspalling marks. It was possible to determine a contact fatigue limit of 3 GPa. During rolling-sliding contact fatigue tests (RSCF), a clearly different behaviour between the two specimens in contact has been observed: the driver shows circumferential and inclined cracks and only inclined cracks appear in the follower. This behaviour can be understood if the effect of the residual stress state in near-surface layers is considered. Before RCF tests, the residual stresses are compressive in all near-surface layers. After RCF tests, strong residual stress relaxation and even reversing behaviour was observed in the axial direction, which facilitates the surface crack initiation in the circumferential direction at rolling track borders.

**Keywords:** rolling contact fatigue; nitriding; X-ray diffraction; residual stresses; fatigue failure mechanisms

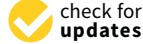

## 1. Introduction

Rolling contact fatigue failure phenomena affect the service life of several different mechanisms and mechanical components, such as, bearings, gears, cams, etc. These kinds of fatigue failure phenomena are usually divided in two main mechanisms: microcrack nucleation and crack propagation [1]. Several damage mechanisms related to the failure of lubricated hertzian contacts have been identified [2–9]. Tallian [10] proposed two deterioration levels (spalling and surface distress) and three kinds of damage mechanisms (surface origin spalling, microspalling or micropitting and subsurface origin spalling). Concerning the damage mechanisms, surface origin spalling seems to be due to high shear stresses which occur during elastohydrodynamic lubrication, mixed or limit lubrication regimes [11]. Their in-depth propagation occurs due to microspalling coalescence, implying high enough tensile loadings to originate stress concentration at cracks tips and lubricant action [10]. Microspalling often arises from the grinding grooves, being the dimensions of this kind of damage of about 7 to 10 μm at the near surface and between 10 to 50 μm in depth. These dimensions can seem very small, but it should be noted that microspalling can grow through the whole contact track, leading to disturbances of the contact condition, originating large spalling and, finally, leading to a complete rupture of the structure [1,12]. On the rolling-sliding conditions, the damaged surface presents V shape microspalling and cross cracks at the track's surface. Cracks presenting an inclination angle to the surface of 20–30 degrees have been observed. Subsurface origin spalling in nitrided steel is generally denominated by "river pattern", due to its similarity when looked from the top, presenting straight cracks, well-delineated, which appear in the lateral zone of the rolling track [13].

Crack nucleation begins at a certain depth, leading to spalling occurrence by axial and circumferential propagation. On the other hand, due to "river pattern" spalling shape and the analysis of the spalling's bottom, some authors suggest that crack nucleation can begin at rolling track edges, spreading out later. This spalling morphology of fragile fracture kind was also observed in carburized steel with no differences between the spalling kind originated by pure rolling and that originated by rolling-sliding one [14]. Continuous demands for better material performance, in applications with highly stressed surface layers due to high contact and fatigue loading, imply development of new materials, new processes and new treatments to be applied to these surface layers [15–19]. New coating materials or surface treatments can be applied to bulk materials to improve the strength of their near surface layers. In the case of steel components, the recent processes and treatments allow us to obtain steels with high purity and without inclusions, implying a better knowledge of the nucleation processes on the role of the fatigue contact phenomena [20]. The trend in recent years has been to research and apply treatments and techniques to the surface layers, that create high compressive residual stresses and gradients of mechanical properties to obtain a high mechanical strength in the zones more requested, keeping the traditional properties of the bulk materials unchanged. The nitride steel studied in this work exhibits important properties in terms of hardness, hot hardness, high toughness, contact fatigue life, wear, and corrosion endurance. These characteristics make it particularly suitable for severe application conditions, namely, aeronautic bearings [21,22], helicopter transmission gears [23], manufacturing specific parts submitted to important efforts [24] and applications to wood machining [25,26]. An experimental and numerical rolling contact study on this quenching and tempered 32 CrMoV 13 steel was made [27]. This paper intends to contribute to a better knowledge on these phenomena, in the nitrided layers, presenting an experimental characterization of the rolling and rolling-sliding contact fatigue failure mechanisms and the fatigue life of the 32 CrMoV 13 nitrided steel, especially in applications of rolling bearings that are subjected to very high contact stresses. In the near future, it is intended to study this phenomenon in nitrided layers of bigger depth and carry out a numerical simulation.

## 2. Materials and Methods

The bulk material employed in the experiments was the quenching and tempered 32 CrMoV 13 steel, whose composition is (wt %): 0.32% C; 0.35% Si; 0.52% Mn; 0.14% Ni; 3.0% Cr; 0.83% Mo; 0.28% V. Austenitizing was performed at 950 °C for 30 min followed by quenching in oil. In the following, the specimens were subjected to tempering, at 635 °C for a holding time of 1 h, followed by cooling in air. Finally, the specimens were subjected to plasma nitriding with parameters chosen to obtain a nitrided layer of about 0.3 mm depth. The thickness, morphology, composition, and structure of the nitrided layers were observed by optical microscopy, scanning electron microscopy (SEM) and X-ray diffraction (XRD). Vickers microhardness profiles were carried out along the treated layers on the cross section of the specimens, using a load of 25 g for 15 s, enabling us to analyse how mechanical properties vary throughout the treated layers.

A two-disc type RCF testing machine with lubrication was used to simulate rolling contact conditions [27]. Specimens of 70 mm diameter, 7 mm width and 40 mm curvature radius were used. In the RCF tests, both discs were subjected to 11.7 m/s linear velocity and the oil temperature was maintained at 56 °C. In RSCF the linear velocity of the discs was $U_1 = 11.7$ m/s and $U_2 = 10.3$ m/s, which means that the faster (driver) specimen has the same rolling and sliding directions and the slower (follower) specimen has opposite rolling and sliding directions. As lubricant, a hydraulic oil with 46 mm$^2$/s kinematic viscosity at 40 °C (ISO VG 46) and a viscosity pressure coefficient equal to 13.39 GPa$^{-1}$ was used [28]. The initial contact conditions and the results obtained for the minimum oil film thickness, *h*, superficial roughness, *Ra*, lambda ratio, Λ, and sliding rate are shown in Table 1. The oil film thickness, *h*, was determined by the formula of Hamrock and Dowson [29]. The

lambda ratio, $\Lambda$, is the relationship between the oil film thickness, $h$, and the composite surface roughness, $\sigma$ [30]:

$$\Lambda = h/\sigma \tag{1}$$

In the RSCF tests the sliding rate, $\lambda$ was determined by the following equation,

$$\lambda = (U_1 - U_2)/(U_1 + U_2) \tag{2}$$

where $U_1 = 11.7$ m/s and $U_2 = 10.3$ m/s are the circumferential linear velocity of the specimens.

**Table 1.** Contact fatigue tests conditions.

| Load [N] | Hertz Pressure, $P_H$ [GPa] | Contact Area [mm] | Minimum Film Thickness, h [µm] | Roughness, Ra [µm] | Lambda Ratio $\Lambda$ | Sliding Rate, $\lambda$ [%] |
|---|---|---|---|---|---|---|
| 960 | 1.9 | 0.467 × 0.511 | 0.403 | 0.72 | 0.407 | 0 |
| 3660 | 3.0 | 0.730 × 0.798 | 0.277 | 0.2 | 0.979 | 6.4 |
| 4000 | 3.1 | 0.752 × 0.822 | 0.363 | 0.7 | 0.367 | 0 |
| 5000 | 3.3 | 0.810 × 0.886 | 0.358 | 0.2 | 1.266 | 0 |
| | | | 0.358 | 0.7 | 0.362 | 0 |
| | | | 0.271 | 0.2 | 0.958 | 6.4 |
| 7000 | 3.7 | 0.906 × 0.991 | 0.349 | 0.2 | 1.234 | 0 |
| | | | 0.265 | | 0.937 | 6.4 |

To identify the failure mechanisms, the specimens were analysed after testing with magnifying lenses, by optical microscopy and SEM. A cut-off of 0.8 mm was used for measuring the arithmetic average value of filtered roughness (Ra) before and after the RCF tests, following the standard DIN 4768. Residual stress determination was performed by XRD using the $\sin^2\psi$ method [31]. Lattice deformations of the {211} diffraction planes of the $\alpha$-Fe phase were determined for six $\phi$ directions using Cr $K_\alpha$ wavelength with vanadium filter in the diffraction beam. Surface and in-depth residual stresses were determined before and after RFC tests, in three zones of the cylindrical specimens, A, B and C, equally spaced by 120 degrees (Figure 1a). Before the RFC tests, the residual stress measurements were also carried out, for some specimens, in three points equal spaced along the width of the specimens, as shown in Figure 1b, to understand the homogeneity of the nitriding treatment. After RCF tests, due to the width of the rolling track, the measures are taken only in the centre of the specimens. The subsurface residual stress was determined at varying depths below the surface, using the electropolishing removal technique. Stress directions were defined, as shown in Figure 1b: $\sigma_{11}$ is the circumferential or hoop stress tangent to the rolling track in the rolling direction, and $\sigma_{22}$ is the axial stress, perpendicular to the rolling direction (Figure 1b). The acquisitions were made for 11 $\psi$, with oscillations $\pm 2°$ in $\phi$, using an exposure time of 50–60 s per peak, for each $\phi$ direction. The X-ray spot diameter was about 1 mm and the penetration depth was 5 µm. The XRD peak position was determined by the centred-centroid method.

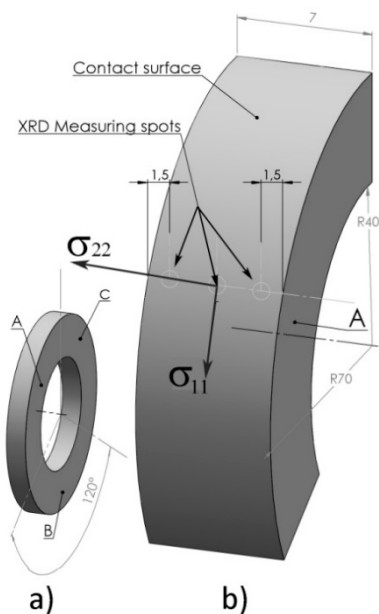

**Figure 1.** Residual stresses measuring spots at the specimen surface. (**a**) Three zones A, B and C at 120 degrees and (**b**) XRD spots equally spaced along the width of the specimens. The central point shows the circumferential or hoop residual stress ($\sigma_{11}$) and axial residual stresses ($\sigma_{22}$).

## 3. Results and Discussion

### 3.1. Mechanical Characterisation of Nitrided Surface Layers

Figure 2a presents the optical micrograph of the cross-section of the nitrided specimens. According to the XRD results, the white layer (also called compound layer) presents, mainly, $\gamma'$ nitrides ($Fe_4N$) and only vestiges of $\varepsilon$ nitrides ($Fe_{2,3}N$). The white layer is 18 $\mu$m thick and the diffusion zone has about 380 $\mu$m depth (according to DIN 50190). Microstructure detail of the white layer surface, obtained by SEM, is presented in Figure 2b).

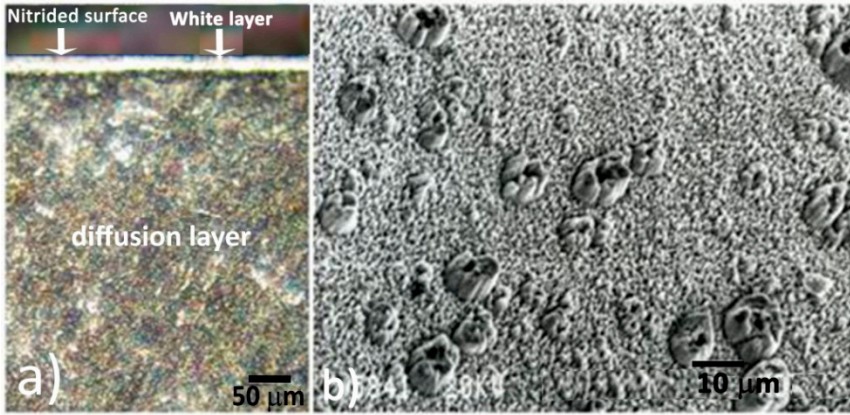

**Figure 2.** 32 CrMoV 13 nitrided steel sample: (**a**) Optical micrograph of cross-section; (**b**) SEM micrograph of the white layer surface.

As can be seen in Figure 3, Vickers microhardness continuously increases from the bulk material to the nitride top surface of the specimens, where maximum values around 800 HV were found, twice those obtained for the bulk material. An affected depth of about 400 $\mu$m is also observed. It means that the surface layers were strongly work hardened due to the nitriding treatment.

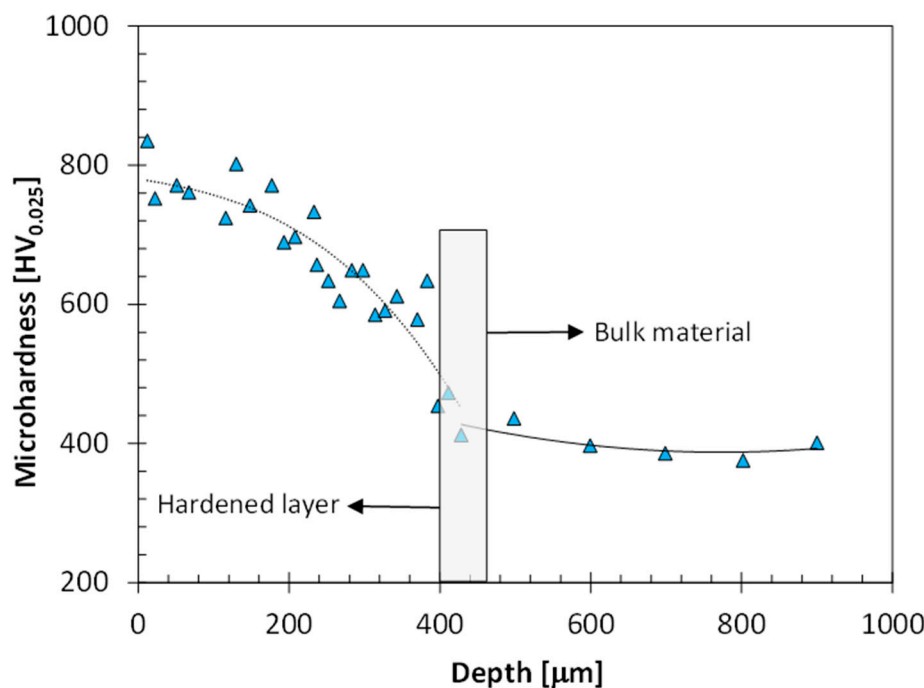

**Figure 3.** In-depth Vickers microhardness depth profile of nitrided specimens.

The residual stresses results at the specimen surface before and after RCF tests are presented in Table 2. Table 3 shows the residual stresses results obtained for specimen A in three points equally spaced along the width of the contact surface (Figure 1b) before RCF tests. The in-depth residual stress distribution at the central point of contact, before (specimen B) and after RCF tests (specimen A), is presented in Figures 4 and 5, respectively. Tables 2 and 3 show that, before RCF tests, the initial residual stresses at the material surface are generally highly compressive due to nitriding treatment. Some scattering on the residual stress values can be observed between each specimen and between different track zones on each specimen.

**Table 2.** Residual stresses at the material surface, before and after RCF tests (before, specimen B, after, specimen A), determined at the central point in three zones A, B and C according to Figure 1.

| Values in [MPa] | | Zone A | | Zone B | | Zone C | |
|---|---|---|---|---|---|---|---|
| | | **Before RCF** | **After RCF** | **Before RCF** | **After RCF** | **Before RCF** | **After RCF** |
| Specimen B | $\sigma_{11}$ | $-386 \pm 61$ | | $-543 \pm 12$ | | $-387 \pm 19$ | |
| | $\sigma_{12}$ | $-430 \pm 62$ | | $-565 \pm 13$ | | $-391 \pm 20$ | |
| Specimen A | $\sigma_{11}$ | $-510 \pm 23$ | $-465 \pm 21$ | $-540 \pm 21$ | $-414 \pm 21$ | $-495 \pm 20$ | $-417 \pm 23$ |
| | $\sigma_{12}$ | $-513 \pm 24$ | $-176 \pm 24$ | $-572 \pm 21$ | $+21 \pm 24$ | $-450 \pm 20$ | $-58 \pm 23$ |

**Table 3.** Residual stresses in the surface of specimen A before RCF tests, zone B, in the three points according to Figure 1b.

| Contact Surface | | Point 1 | Central Point | Point 2 |
|---|---|---|---|---|
| Specimen A | $\sigma_{11}$ | $-502 \pm 26$ | $-540 \pm 21$ | $-502 \pm 22$ |
| | $\sigma_{12}$ | $-484 \pm 28$ | $-572 \pm 21$ | $-416 \pm 23$ |

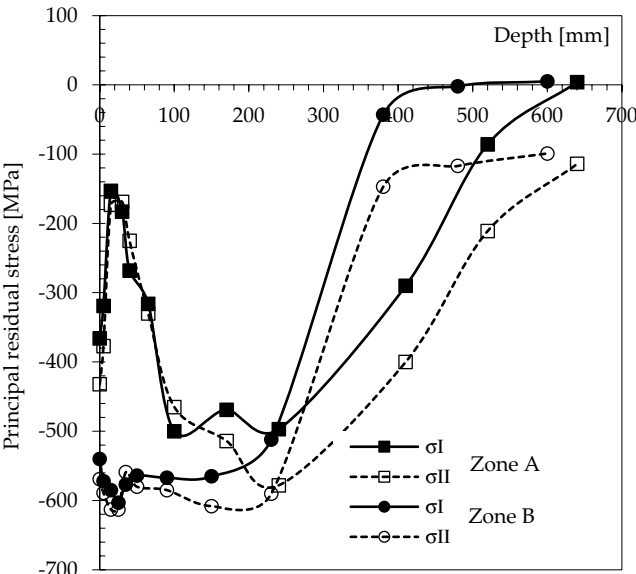

**Figure 4.** In-depth residual stress profiles, before RCF tests, in the central point for zones A and B of specimen B.

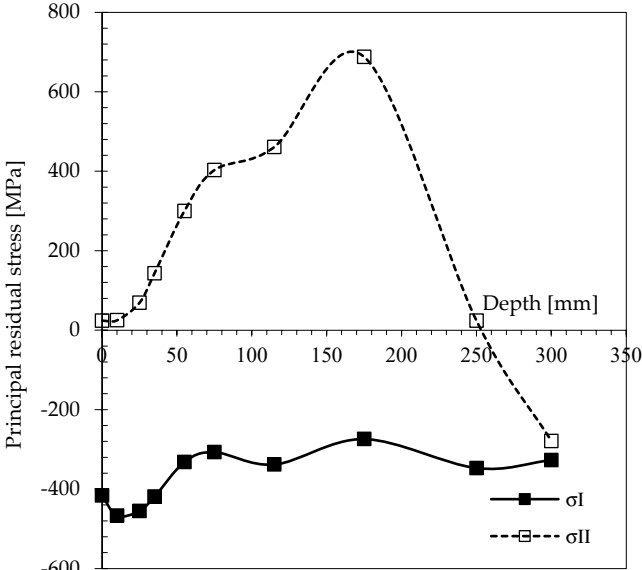

**Figure 5.** In-depth residual stress profile after RCF tests ($P_H$ = 3.1 GPa, N = 6 × 10$^4$ cycles, Λ = 0.367), of specimen A.

The measurements in three points throughout the track width (Table 3) show that the compressive residual stresses at the centre of the contact surface are a little higher than those obtained at edges near zones. However, the results reveal a good homogeneity of the nitriding treatment in the width of the specimens. After RCF tests, it was observed that there was no substantial change in the compressive residual stress in the circumferential direction. However, in the axial direction a strong relieving was observed, including reversing to tensile stress in several cases (such as in specimen A, as shown in Figure 5).

Figure 4 shows the in-depth residual stress profiles obtained at two different zones A and B of specimen B, before RCF tests. In-depth compressive residual stress distribution, induced by the nitriding surface treatment, before RCF tests, presents a maximum compressive stress value between −500 and −600 MPa at a depth of about 200 μm, followed by a decrease of the compressive residual stresses up to a depth of about 400 to 600 μm.

The in-depth residual stress distribution, after RCF, in specimen A, can be observed in Figure 5. Low stress relaxation in the longitudinal direction after the RCF tests, expressed by the circumferential compressive residual stresses ($\sigma_{11}$), is observed, as also reported by Marion Le et al. [32]. These stresses are relieved near the surface and present a constant value around −400 MPa in the first 300 μm depth. However, we should emphasize the tensile behaviour of the axial residual stress ($\sigma_{22}$), throughout a depth of ~250 μm, induced by the rolling contact loading.

Comparing Figures 4 and 5, it is observed that residual stress in the axial direction reverses its sign after the RCF tests, changing from compressive to tensile. A maximum tensile stress value of approximately +680 MPa occurs at ~180 μm depth. The explanation for the appearance of such high tensile stresses is not clear and not fully understood. This behaviour may be explained by the plastic strain heterogeneities induced by the cyclic Hertz contact stresses, coupled with the characteristics of the nitrided layers. Coules et al. [33] performed a numerical simulation of the rolling process, in a locally-rolled aluminium alloy specimen, using finite element analysis, in order to predict the residual stress field and to study the pattern of plasticity, which occurs in the specimen during and after rolling. The residual stress prediction of their FEM model was compared with an experimentally obtained one using neutron diffraction. According to the obtained results, interesting findings were reported. This included the appearance of tensile residual stresses in the transverse direction (axial direction) of the samples, in particular at border and ends of the rolling tracks, but also at their centres, even of lower magnitudes. In any case, tensile residual stress normal to the plane of a crack-like defect is highly beneficial for crack propagation (stress ratio increase). Therefore, the presence of high tensile residual stresses is highly undesirable. Thus, in this case, appearance of longitudinal cracks leading to fatigue damage could be of high concern during contact fatigue of nitrided components.

### 3.2. Failure Mechanisms in RCF Tests

The S-N curves and the fracture images of some RCF tests are show in Figure 6 where the arrows mean that the specimen did not fail, that is, it reached at least 10 million cycles. There are two groups of fractured specimens: one with a high value of Hertz pressure ($P_H > 3$ GPa) and short life regime ($<10^5$) and the other with low Hertz pressure ($P_H \leq 3$ GPa) and long-life regime ($>10^6$ cycles). Therefore, it seems that a contact fatigue of about 3 GPa exists, for the 32 CrMoV 13 nitrided steel studied. Figure 7 shows the surface's micrographs corresponding to the second mentioned behaviour ($P_H \leq 3$ GPa).

The rolling contact track surface can be seen in the Figure 7a). Two magnifications of this rolling track show a flattening of the white layer (Figure 7b) and the beginning of a superficial circumferential crack (Figure 7c). All specimens subjected to a Hertz pressure below the referred contact fatigue limit (3 GPa) have presented a similar track morphology and very high service life.

When the Hertz pressure increases, the contact stress exceeds the contact fatigue limit, and a distinct behaviour appears. Specimens present a short life, and the crack nucleation happens at the surface at the border of the rolling contact surface. This is probably caused by the tensile stress that is expected in this region, in accordance with the results of Fernández-Valdés [7] and Coules [33]. At the surface, the crack propagates in the circumferential direction (rolling direction, Figure 8a), probably related to the tensile stresses, while in depth it grows perpendicular to the rolling track (Figure 8b,c). After a given small depth, cracks bend almost 45 degrees (Figure 8c) in a direction to the rolling track centre, probably due to the action of the shear stress and the material's softening induced by the plastic deformation. The spalling phenomenon occurs when two or more cracks find each other, or a critical strength area is attained, which is a characteristic brittle fracture of the nitrided surfaces.

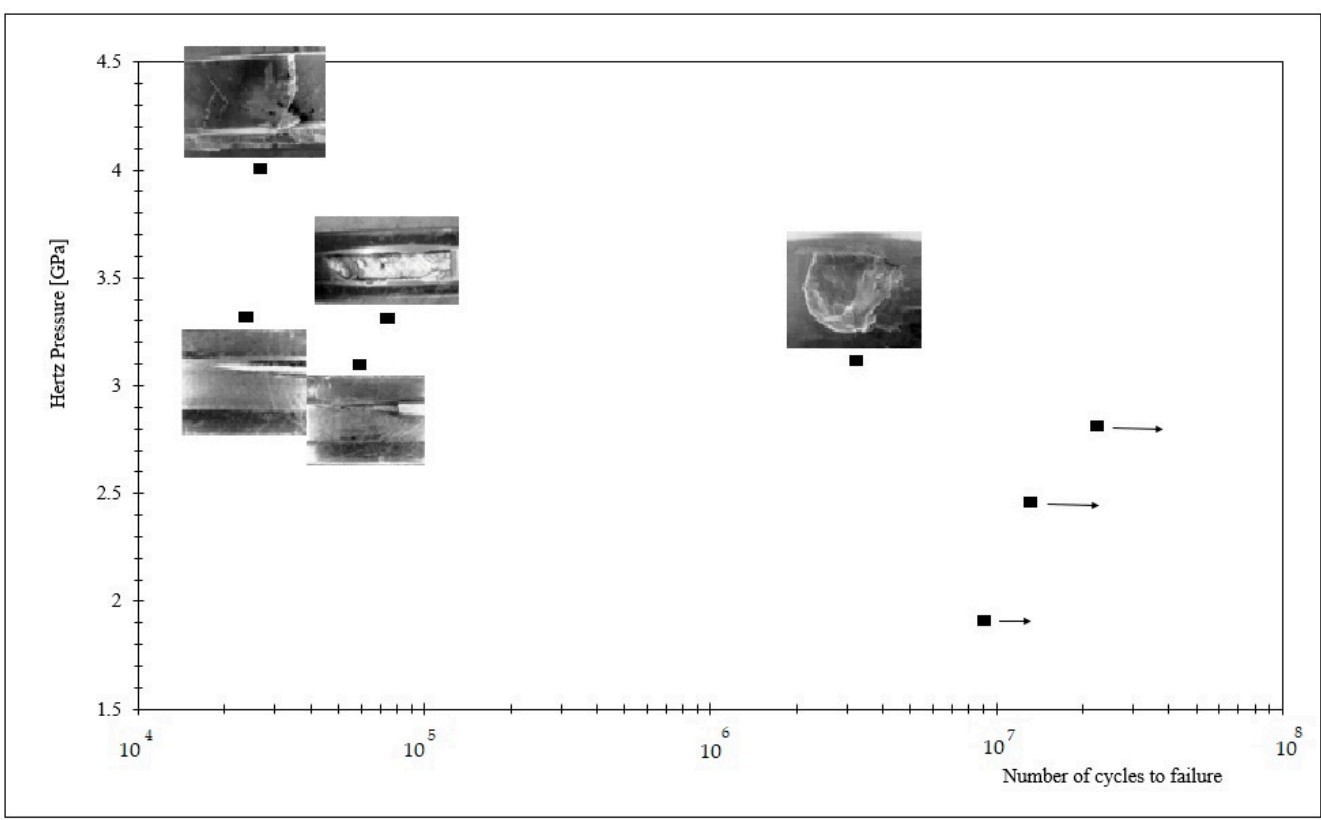

**Figure 6.** S-N curve and fracture images of the nitrided specimens in RCF tests.

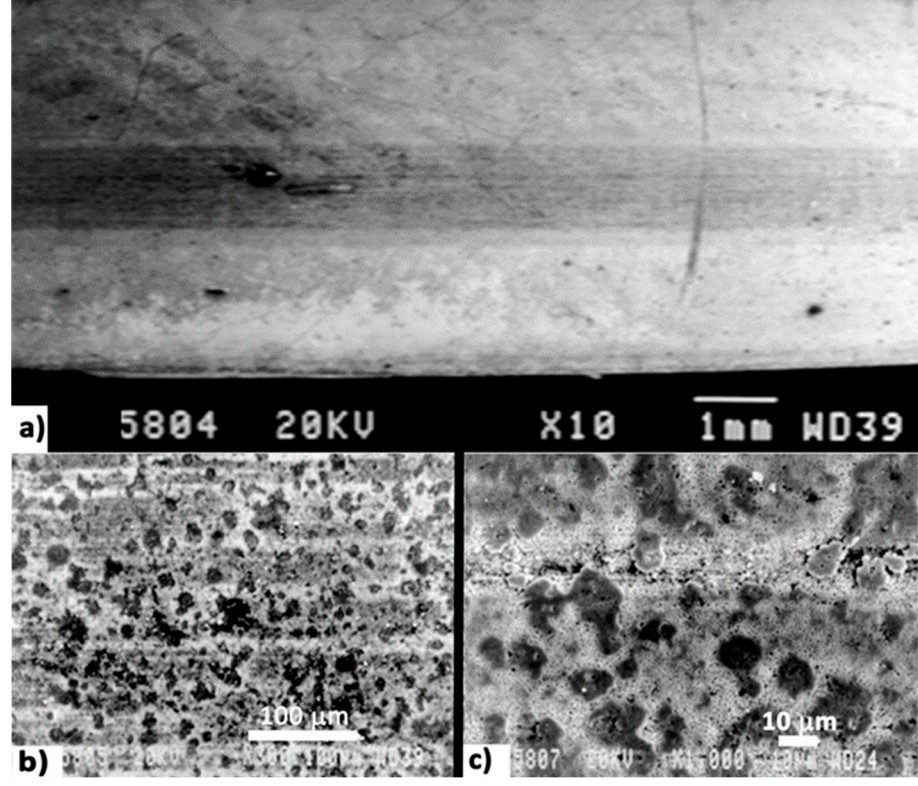

**Figure 7.** (**a**) Rolling track's SEM micrographs after RCF tests ($P_H$ = 1.9 GPa; N = 9 × $10^6$ cycles; $\Lambda$ = 0.407); (**b**) flattening of the white layer (300×); (**c**) beginning of superficial circumferential crack (1000×).

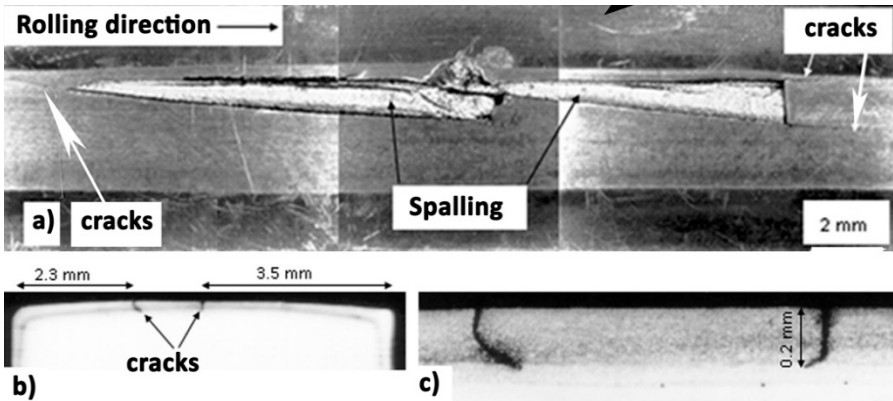

**Figure 8.** Surface morphology after RCF tests. (**a**) Circumferential cracks and spalling, (**b**) transversal cracks in the cross-section and (**c**) zoom of (**b**) ($P_H$ = 3.3 GPa; N = 2.43 × 10⁴ cycles; Λ = 0.362).

To better understand the evolution of this process, one test was continued, after crack detection, until the spall appearance. Figure 9 shows optical micrographs of the rolling contact surface track of this specimen tested at $P_H$ = 3.7 GPa and 1.8 × 10⁵ loading cycles.

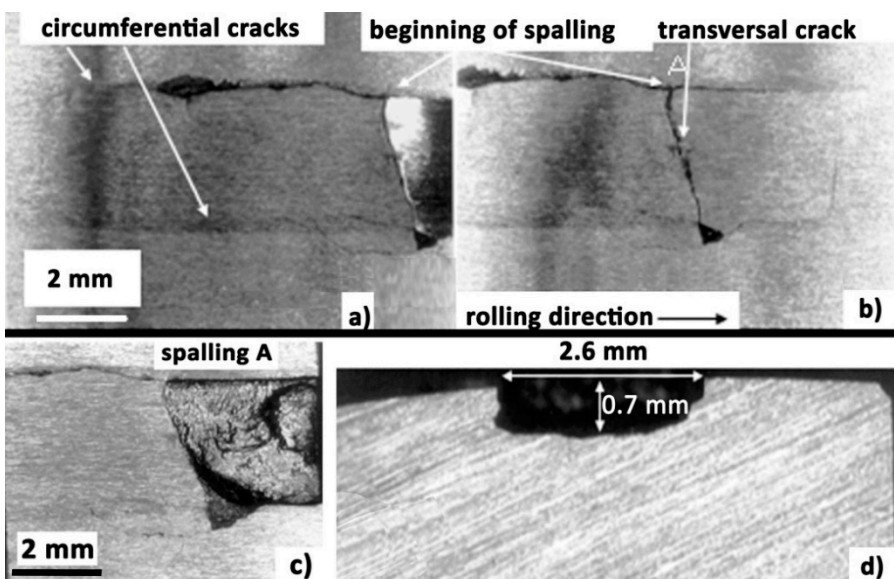

**Figure 9.** Morphology of the rolling contact surface after RCF test ($P_H$ = 3.7 GPa; N = 1.8 × 10⁵ cycles; Λ = 1.234). (**a**) Circumferential cracks at the borders of the rolling contact track; (**b**) circumferential and transversal cracks at the point A, in the rolling contact surface, before the spalling phenomenon appearance shown in (**c**); (**d**) cross section of the spalling shown in (**c**).

The observation of Figure 9 leads to the conclusion that the spalling phenomenon can be described in three stages. A first stage, characterised by the circumferential surface crack nucleation at rolling contact track border (Figure 9a), where the appearance of tensile residual stress in the axial/transverse direction should potentiate the process, is followed by a second stage where, simultaneously, cracks propagate in circumferential and in-depth directions (Figure 9b). The in-depth propagation near the surface, i.e., in the hardened layer, occurs perpendicularly to the surface, which is a typical feature of brittle fracture behaviour occurring in planes perpendicular to the maximum principal stress. A third stage begins when a given critical depth is attained. This stage is probably related to the depth where the maximum hertzian shear stress occurs (which for this contact conditions is 1180 MPa at a depth of 0.453 mm) and with the softening material due to the plastic deformation, where the cracks bend in the direction of the rolling track centre, with a slope of ≈45 degrees as

it occurs in ductile materials in planes where the shear stress attains a maximum. This is a typical feature of the materials presenting a ductile fracture behaviour. After this stage, when the crack attains a critical length, the remaining area does not support the contact pressure and the material is pulled up, which is a well-known characteristic of spalling phenomena (Figure 9c,d). Note that the spalling damage at the rolling track is delimited by transversal cracks, which will be linked to the existing circumferential ones Figure 9b).

### 3.3. Failure Mechanisms in RSCF

Beyond the rolling contact fatigue tests, some specimens were subjected to rolling-sliding contact fatigue tests (RSCF). The main goal was to identify the damage mechanisms that occur in these contact conditions in the driver (where the rolling and sliding directions are the same) and follower (where the rolling and sliding directions are opposite) specimens. Two kinds of damage morphology, depending on applied loading, has been verified. For high Hertz contact pressures and fast linear contact velocities, spalling and circumferential cracks were the main damage mechanisms. For lower Hertz contact pressure, no cracking was observed and only microspalling appears, increasing for the slower specimens (follower) due to different rolling and sliding directions. In the driver specimens two kinds of surface cracks exist, appearing side by side at rolling track borders: circumferential cracks and inclined cracks, as Figure 10a,b shows.

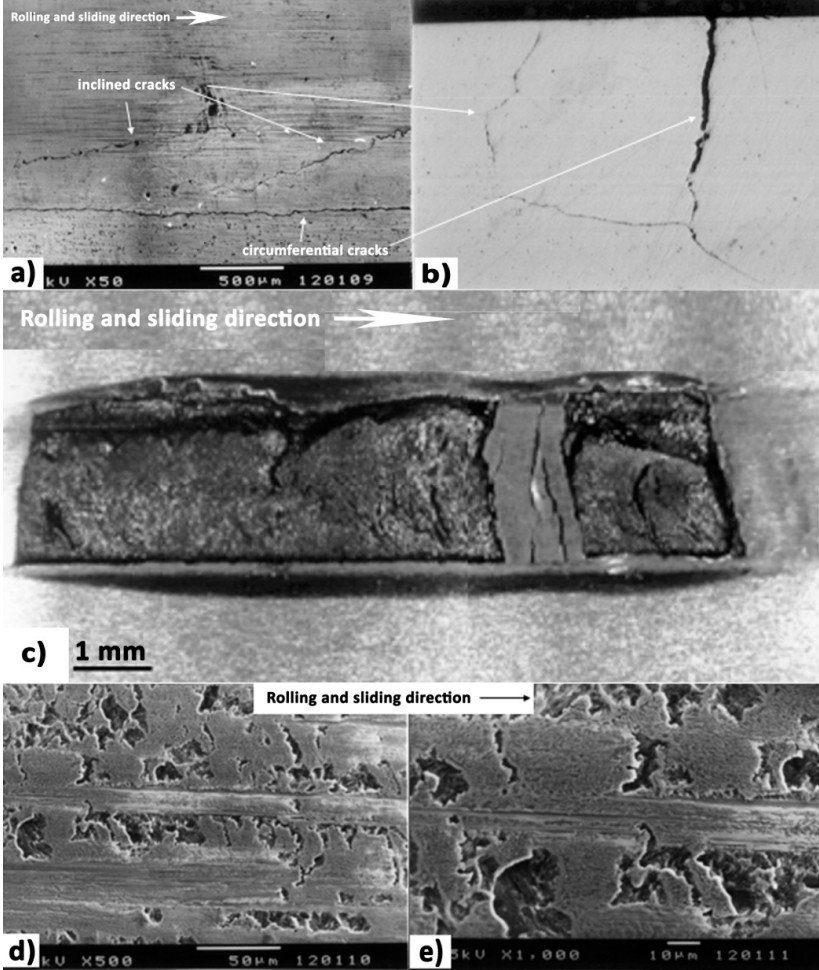

**Figure 10.** Surface of faster specimen (driver) after RSCF tests ($P_H$ = 3.7 GPa; N = 3 × 10^4 cycles; Λ = 0.937; λ = 6.4%). (**a**) SEM micrographs of surface circumferential and inclined cracks; (**b**) cross-sectional optical micrograph showing circumferential and inclined cracks; (**c**) surface double spalling; (**d**) microspalling in the rolling track; (**e**) microspalling in the rolling track.

Circumferential cracks in the rolling track are bigger and deeper than the inclined ones (Figure 10b), which are outside of circumferential cracks and propagate to out of the rolling track. As consequence, in the damaged zone, two spallings, separated by a little distance, can be seen in Figure 10c. Spallings present a rectangular shape and the smaller side covers the whole rolling track width, being one of the greater sides perfectly rectilinear and its opposite side slightly curved. Microspalling can also be seen in the contact surface of driver specimens, as Figure 10d,e shows. Damage mechanisms in the follower specimens are different than those already described for the driver specimens. In this case, no circumferential cracks have been detected and only inclined cracks appear. However, the initiation crack occurs clearly inside of the rolling track, although at the end its propagation spreads to out of the rolling track, contrary to the observed behaviour in the driver specimens, where inclined cracks are initiated at rolling track borders and propagate to out of the rolling track. The surface and the cross section of these specimens can be seen in Figure 11.

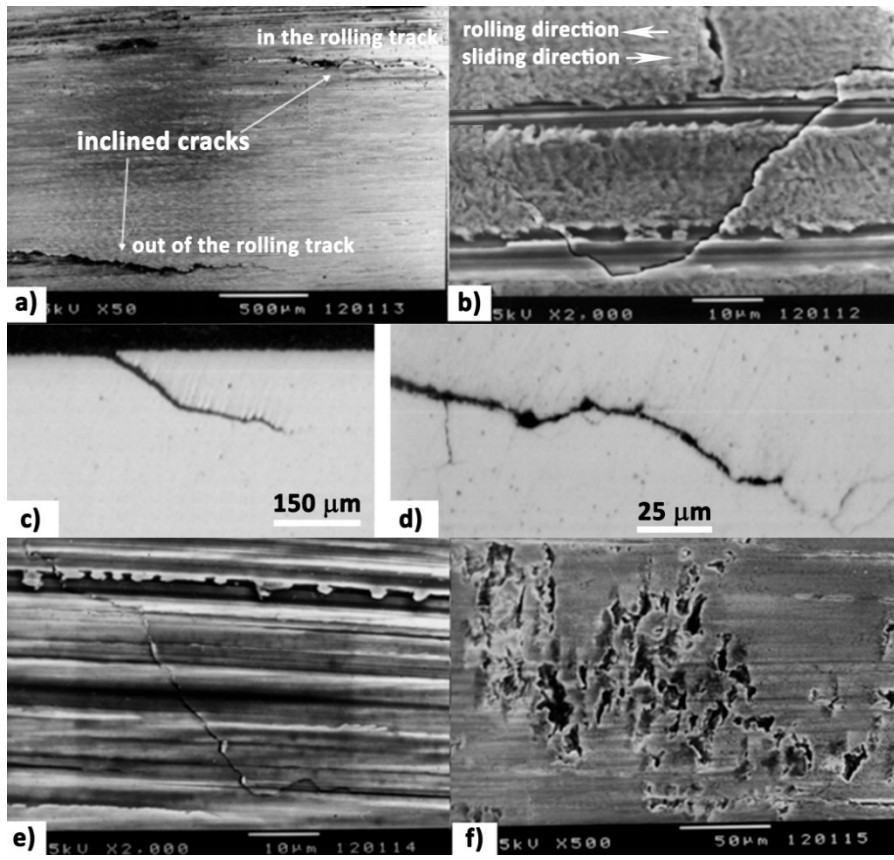

**Figure 11.** SEM and optical micrographs of superficial inclined cracks in a slower (follower) specimen after RSCF tests ($P_H$ = 3.7 GPa; N = 1.8 × $10^5$ cycles; $\Lambda$ = 0.958; $\lambda$ = 6.4%); (**a**) the upper crack appeared inside the rolling track and the bottom crack appeared outside the rolling track; (**b**) crack nucleation inside the rolling track; (**c**) optical cross-section micrograph of the inclined crack; (**d**) zoom of the front crack showed in (**c**) and corresponding secondary cracks; (**e**) superficial inclined crack out of the rolling track; (**f**) microspallings in the rolling track.

The inclined surface cracks in the follower specimens show a different behaviour when compared to the driver specimens. These cracks can appear inside of the rolling track (Figure 11a) as well as outside of the rolling track (Figure 11a,e). These cracks are also inclined through the depth (Figure 11c,d). Indeed, for driver specimens, the in-depth propagation happens perpendicularly to the rolling direction (Figure 10b). The surface

shear stresses induced by the rolling-sliding contact should be responsible for this observed behaviour. Multiple microspallings have also been observed, as shown in Figure 11f.

For Hertz contact pressure equal or lower than 3 GPa, only microspallings have been observed. Not enough damage occurred to originate cracks or spalls. The results are shown in Figure 12. In the driver specimens, most of the observed microspallings seem to have originated in the grooves of the rolling track due to the fine grinding procedure imposed on the specimens after the heat treatment of nitriding (Figure 12a,b). In the follower specimens, the microspallings appeared in any part of the rolling track, as seen in Figure 12c,d.

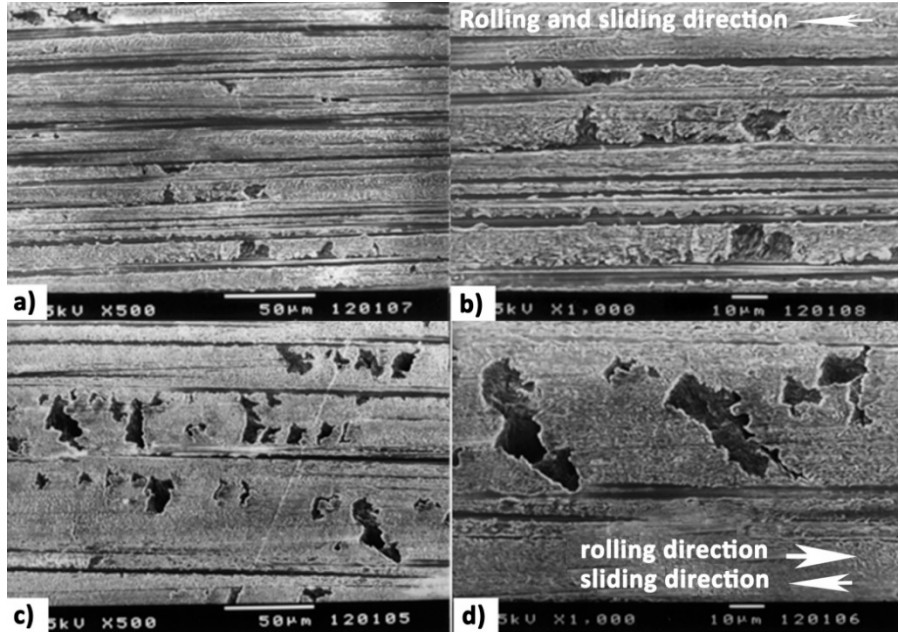

**Figure 12.** SEM micrographs after RSCF tests ($P_H$ = 3 GPa; N = 9.9x10$^6$ cycles; $\Lambda$ = 0.979, $\lambda$ = 6.4%). (**a**,**b**) Microspallings and grooves in the rolling track of the faster specimens (driver); (**c**,**d**) microspallings and grooves in the rolling track of the slower specimens (follower).

## 4. Conclusions

Fatigue failure mechanisms on the 32 CrMoV 13 nitrided steel subjected to rolling and rolling-sliding contact fatigue were investigated and characterized. Although some scattering has been observed, the heat treatment of nitriding clearly induced compressive residual stresses, with an observed maximum of −600 MPa and an affected surface layer of about 400 μm depth.

After RCF tests, although in a circumferential direction, the residual stresses remain almost unchanged, and in an axial direction a strong stress relief has been observed, including reversing from compressive to tensile stresses. Maximum values of +600 MPa at 180–200 μm depth have been observed.

In RCF tests, we found a contact fatigue limit corresponding to a Hertz pressure of 3 GPa. In this case, the main damage mechanism was the circumferential cracking appearance at the contact surface. These cracks can appear in one or both sides of the rolling track border, thus affecting the service life of the specimens. The tensile axial residual stresses contribute to the nucleation of these circumferential cracks, its consequent propagation and final spalling appearance. Cracking and spalling mechanisms can be described in three stages: firstly, the crack nucleation occurs at the rolling track border by a combined action of contact Hertz loading and tensile axial residual stresses; secondly, these cracks propagate, simultaneously, in the circumferential and in-depth direction, perpendicularly to the rolling direction; thirdly, at 200–300 μm depth, the cracks bend to the interior, i.e., in the direction of the specimen's centre, probably due to the effect

of maximum shear stress and/or a softening material due to plastic deformation until a critical crack length is attained which can lead to a spalling.

In RSCF tests, depending on the magnitude of the Hertz pressure, two situations have been observed: above 3 GPa, in the faster rolling specimens, where the rolling and sliding directions are the same, a similar damage mechanism already seen during the pure RCF tests, i.e., surface circumferential cracking nucleation in the rolling track borders. In the slower specimens, where the rolling and sliding directions are opposite, inclined circumferential cracks appear at centre of the rolling track and propagate at low speed (comparatively to the faster specimens). Below 3 GPa, no spalling has been observed and only microspalling has been observed, mainly in the slowest specimens.

**Author Contributions:** Conceptualization, L.C., A.C.B., J.P.N., M.J.M.; Investigation, L.C., A.C.B.; writing—original draft preparation, L.C., J.P.N.; writing—review and editing, L.C., A.C.B., J.P.N., M.J.M. All authors have read and agreed to the published version of the manuscript.

**Funding:** This research received no external funding.

**Institutional Review Board Statement:** Not applicable.

**Informed Consent Statement:** Not applicable.

**Acknowledgments:** This work was supported by national funds from FCT—Fundação para a Ciência e a Tecnologia, I.P., within the projects UIDB/04564/2020 and UIDP/04564/2020.

**Conflicts of Interest:** The authors declare no conflict of interest.

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
