# Peer review of "Rolling and Rolling-Sliding Contact Fatigue Failure Mechanisms in 32 CrMoV 13 Nitrided Steel—An Experimental Study"

_applsci, doi:10.3390/app112110499_

Round 1
Reviewer 1 Report
This paper presents research on crack nucleation and propagation in rolling and rolling-sliding contact fatigue of 32 CrMoV 13 nitrided steel, the study being interesting and useful to researchers and reliability design engineers.
I consider that the paper brings new elements to RCF knowledge, especially on the influence of the hoop stress and slide-to-roll ratio on the propagation of the cracks.
There are some minor points that the authors should clarify prior to the publication of the article:
- The novelty of the paper is not clearly stated, as the authors just mentioned that there is some published research [28-32] on the classical fatigue strength of nitrided layers. One or two phrases should summarize the findings of the previously published research, that is [28] to [32]. Also, the originality of the current article must be assessed.
- The authors mentioned that there were three kinematic cases: pure rolling, rolling with sliding in the same direction, and rolling and sliding in the opposite directions. Also, it is specified that there is a drive roller and a follower. In my opinion, Table 1 does not present all the carried out experiments, as I expect at least two slide-to-roll ratios, excepting the nill value.
- The samples have a 40 mm curvature. In this case, why in Figure 1 there is no curvature in the axial direction and how did the authors choose the XRD measuring spots?! The results should be influenced by the presence of the curvature and uneven contact area between the rollers.
- Below Table 2, the authors should specify what specimen 1.3, 2.3, etc. mean. Some readers could get confused.
- In line 266, an explanation is necessary regarding the increased number of micro-spalling at a lower speed.
Reviewer 2 Report
The authors reported the characterization results of rolling/rolling-sliding contact fatigue failure mechanisms on the 32CrMoV13 nitrided steel by using multiple techniques including optical microscopy, SEM, XRD. The authors appear to have been studying the same alloy system for many years.
- The authors gave an intensive literature review in the “Introduction” section; however, it is not very clear on the reason(s) that the authors selected this 32CrMoV13 alloy for the current study. Also, it is not clear what related studies have been done on this alloy and what need to be addressed. It is not enough for the audience to understand the importance/essential of the presented research with only stating what the manuscript intends to do. More details about the research background of contact fatigue failure mechanisms related to this 32CrMoV13 alloy are suggested.
- 2b. What are those features (~ 10 um in diameter) in the SEM image?
- Page 7, section 3.1. The authors stated “A maximum tensile stress value of approximately +680 MPa occurs at ~180 μm depth. The explanation for the appearance of such high tensile stresses is not clear and not fully understood.” Is this related to the N concentration or nitride phase? Was any EDS done to reveal the composition differences at different depth?
- It would significantly strengthen this research if the authors include simulation on the stress/strain from the rolling/rolling-sliding. This is recommended but not required.
Reviewer 3 Report
Please consider the comments below in the revised draft.
- Figure 6
Figure6 represents the SN diagram based on several experiments.
As the author may know, in the high cycle fatigue area, it can be seen that the smaller the load, the shorter the fatigue life. Please explain clearly about this result. In addition, the reviewer believes that it is difficult to reach the conclusion that the SN diagram has been derived from several experiments. This also needs an explanation. - . Conclusion:
The sentence below needs an errata check.
The reviewer believes that the location of "~ which leads to a spalling." fails to express the meaning of the whole sentence.
: ~ probably due to the effect of maximum shear stress and a softening material due to plastic deformation until a critical crack length is attained which leads to a spalling.
